# Time-Resolved cw Thermal Z-scan for Nanoparticles Scattering Evaluation in Liquid Suspension

**DOI:** 10.3390/ma15145008

**Published:** 2022-07-19

**Authors:** Christophe Cassagne, Oumar Ba, Georges Boudebs

**Affiliations:** Univ Angers, LPHIA, SFR MATRIX, F-49000 Angers, France; christophe.cassagne@univ-angers.fr (C.C.); oumar.ba@etud.univ-angers.fr (O.B.)

**Keywords:** time-resolved Z-scan, thermal lens, Z-scan, linear absorption, scattering efficiency

## Abstract

The thermal lens effect is analyzed as a time-resolved Z-scan measurement using cw-single Gaussian beam configuration. The main characteristics of the measurement method are determined. We focus on the evaluation of the measurement error from statistical calculations to also check the linearity of the response and the way to extract the thermo-optical characteristics of absorbing liquids. The results are also applied to demonstrate the feasibility of absorption and scattering efficiencies determination on gold nanoparticles of 5 and 50 nm diameters.

## 1. Introduction

In many interdisciplinary areas, nonlinear properties of liquids or suspension of liquids became an important tool. The response of a material to low laser radiation is well known: the parameters *n* and *α* respectively defining the linear index and the absorption characterize this effect. However, when the radiation becomes intense, a variation of index and absorption can appear due to several physical phenomena, including thermal effects. The latter depends on the properties of the sample mainly on its one-photon linear absorption, but also the thermal changes in refractive index that could be due to two-photon absorption [1]. Generally, to report on thermal response, Gaussian beam-induced lensing is used. This local change of temperature in turn produces a change in the refractive index that is characterized by the thermo-optical parameter *dn/dT*. Since Gordon et al. [2] reported on this effect, considerable work has been conducted in this field and several studies were carried out on the characterization of thermo-optical behavior of materials [3,4,5]. To estimate the material absorbance, measuring the diffraction of light in single beam or dual-beam methods [6] allow to obtain highly sensitive techniques. Consequently, different physical phenomena can be explored using the thermal lens (TL) principle to study: for example, properties of solid-state laser glasses [7]. In [8], the authors investigated the molecular/particle dynamics and, in [9], by measuring photothermal parameters of opaque solids. Moreover, Z-scan [10] based methods were performed at different regimes of excitation to understand the origin of the third-order nonlinear optical response [11] and the influence of the accumulated thermal effect, which can be an artifact in femtosecond closed-aperture Z-scan measurements [12]. Additionally, they can be used to evaluate optically-induced temperature changes in colloidal samples for photothermal therapy [13]. Already, we have experimentally demonstrated the feasibility of extracting an image of the phase shift induced by TL and applied this method to transversally map an inhomogeneous thin film doped with different concentrations of silver nanoparticles [14]. Additionally, numerical simulation considered the whole system, including both optical and thermal parameters to obtain more reliable temperature estimation in absorbing media whether in time-resolved or steady-state Z-scan techniques [15].

Following the same approach as the work conducted by [15,16], we explore in this paper the thermal behavior of some absorbing liquids with different solutes and colloidal suspensions. Measurements were performed on different concentrations and different liquids to calibrate and optimize the setup. Then we analyze the behavior of two colloidal suspensions of spherical gold nanoparticles of different diameters (5 and 50 nm). We show that it is possible to find out the scattering part of the light during the TL measurement from that purely absorbed by the liquid. The results obtained experimentally are fully consistent with those predicted by the Mie theory. So, the first aim of this paper is to demonstrate the feasibility of characterizing thermal effects through the quantum yield of a radiation-induced process, which is equal to the number of times a given event occurs divided by the number of photons absorbed by the system. These measurements are very useful for the characterization of the molecular fluorophores or plasmonic nanoparticles in solution.

## 2. Theory

Following reference [17], some classical assumptions are made: (i) the thin sample approximation; (ii) the beam waist is considered small when compared to the transverse dimensions of the cell so that the heat conduction through the ends can be neglected, and thus the temperature variation can be taken as purely radial. These approximations are well validated in practice when considering beam waists of 20–30 µm and cells of about $1\times 10\;{\mathrm{cm}}^{2}$, with the transverse section having approximately 1 mm thickness. An expression of the temperature change as a function of radius and time ∆T(r,t) can be obtained by solving the non-steady-state heat equation when a Gaussian beam is illuminating the medium. The following symbols will be used: α absorption coefficient in $m^{-1}$; ρ density in $\mathrm{Kg}/m^{3}$; I beam intensity in $W/m^{2}$; L thickness of the medium (cell) in m; P beam power in W; c specific heat in J/Kg/K; ω beam radius in m; ∆T temperature variation in °K; k thermal conductivity in J/s/m/K; λ wavelength in m. The variation in temperature in the medium, taking into account the aberrant nature of the thermal lens [17,18], is given by:(1)$$∆T\left(r,t\right)=\frac{2P\alpha }{\pi c\rho \omega^{2}}\left(1-\emptyset \right)\int_{0}^{t}\left(\frac{1}{1+2t’/t_{c}}\right)\exp\left(\frac{-2r^{2}/\omega^{2}}{1+2t^{\prime }/t_{c}}\right)dt’,$$
where $t_{c}=c\rho \omega^{2}/4k$ is the characteristic buildup time constant and (1−∅) is a factor accounting for the absorbed energy totally converted to heat [19]. Then, ∅ represents the ratio between the number of photons emitted to the number of photons absorbed and converted to heat. The emission accounts for fluorescence, luminescence and/or part of the light energy which has been scattered by nanoparticles depending on their size.

In a separate paper, we performed the simulation of the temperature distribution in the thermal lens according to Equation (1) in order to validate our numerical measurement approach. The result of the temperature distribution at various times $t/t_{c}$ was presented in Figure 1 in [15] and compared to the beam waist of the exciting Gaussian beam, using the same parameters as those shown in Figure 2 of reference [17]. 

The change in the linear index due to this temperature variation is: (2)$$\Delta n\left(r,t\right)=\frac{dn}{dT}\Delta T\left(r,t\right),$$
with dn/dT denoting the algebraic value of the hermo-optical coefficient, which is frequently negative, and $\Delta n\left(x,t\right)=n\left(r,t\right)-n_{0}$ where $n_{0}$ is the refractive index at the initial temperature. The phase shift is related to Δn as usual, using Δφ=2πΔnL/λ with λ designing the wavelength of the beam inducing and probing the phase in the single beam configuration that we are considering.

The TL schematic diagram of the method used is shown in Figure 1. This setup has the advantage of combining a Z-scan profile when the sample moves along z and a TL signal describing the time-response profile when the chopper rotates for each position z. The signal due to the phase shift is obtained through the beam splitter BS, where a photodiode PD1 acquires the central variation of the diffracted far-field beam intensity. At the same time, the reflection on BS allows to measure through L_2_, on PD2, the open aperture normalized transmittance characterizing the absorption versus the intensity of the tested medium in the cell via a configuration as defined in the Z-scan technique [10]. Further, by using cw laser excitation, in which the power and the transverse mode profile are very stable, problems with incident power fluctuations are minimized when compared to that with pulsed lasers used in third-order nonlinear characterization. For each z position of the cell, the top clock is given by a PD integrated in the chopper Ch and connected to the oscilloscope to trigger the signal (the link is not shown in Figure 1). This photodiode controls the data recording on PD1 and PD2. If we consider the stationary signal obtained on PD1 (after a few tens of ms), and depending on the cell position, a variation of the far-field diffracted intensity will occur depending on the sign of the phase shift induced in the cell (and therefore on the z position). According to the Z-scan normalization, for a negative phase shift, the characteristic curve of the signal profile will be given by a maximum for a pre-focal position followed by a minimum for a post-focal position. Note that with the TL normalization defined hereafter, the characteristic curve signature will be inverted. Here, the refractive signal is defined as being the fractional intensity change when the cell is located at z after t times exposure of the material: S(z,t)=[I(z,t)−I(z,∞)]/I(z,∞). For a photodiode placed very far from the cell, considering a number of approximations, analytical calculations at r=0 for the in-axis far-field intensity variation provided [17]: (3a)$$S\left(z,t\right)=-1+\frac{1-\theta tan^{-1}\left[\frac{2V}{3+V^{2}+\left(9+V^{2}\right)\left(\frac{t_{cz}}{t}\right)}\right]}{1-\theta tan^{-1}\left[\frac{2V}{3+V^{2}}\right]},$$
where
(3b)$$\theta =\frac{PL\alpha \left(1-\emptyset \right)\left(dn/dT\right)}{\lambda k},$$
$V=z/Z_{0}$ with $Z_{0}=\pi \omega_{0f}^{2}/\lambda$ and $\omega_{0f}$ being the focused beam waist at z=0. It is appropriate that θ remains sufficiently small because in most experiments invoking thermal lenses using the established analytical relations, *θ* is of the order of 0.1 or less. Note that, our TL signal is to be taken in different *z* positions, each position presents a different beam radius value and consequently:(4)$$t_{cz}=\frac{c\rho w{\left(z\right)}^{2}}{4k},$$
where $\omega {\left(z\right)}^{2}=\omega_{0f}^{2}\left[1+{\left(z/Z_{0}\right)}^{2}\right].$ So, $t_{cz}=t_{c0}\left[1+{\left(z/Z_{0}\right)}^{2}\right]$ with $t_{c0}=c\rho \omega_{0f}^{2}/4k$ is the characteristic TL signal response time at z=0.

Setting t=0 in Equation (3a) allows to obtain the total fractional intensity change as a function of the normalized position V:(5)$$S\left(V\right)=-1+\frac{1}{1-\theta tan^{-1}\left(\frac{2V}{3+V^{2}}\right)}.$$

The derivative of Equation (5) allows to optimize this signal, finding positions of the sample around $V=\pm \sqrt{3}\;$ where the signal S is extremal. As it was shown in [15], a numerical study of the whole thermal lensing and optical processes to evaluate the accuracy related to the measurement using the simplified analytical relation (3a) and therefore (5) gave only an order of magnitude of the desired coefficients to be measured θ and $t_{c0}$. These relations, generally used to estimate the thermo-optical coefficients, have the advantage of existing, but a preliminary calibration of the signal must be conducted in order to obtain reliable and accurate measurements. Furthermore, it was found experimentally in [16] that the signal is maximal at variable z-positions depending on the final acquisition time of PD1. This logically implies that measurements should be performed around $z\approx \pm \sqrt{3}Z_{0}$ positions at locations with the highest possible signal-to-noise ratio (Sig/N). Another study based on the diffracted energy made with the Z-scan technique for third-order optical nonlinearity measurements showed that the sensitivity and the Sig/N were both maximal for a linear transmittance of the Gaussian beam through closed apertures equal to 0.7 [20,21]. If we consider this optimized value, relations (3) and (5) would, therefore, not be correct because they were established at the center of the diffracted field. So, for a transmittance of the closed aperture around 0, then D should be relatively high when compared to the Rayleigh range of the focused beam waist. 

## 3. Experimental Results

The experimental setup (Figure 1) is composed of a cw Nd:YAG laser emitting at 532 nm, delivering through a polarizing system a variable beam power from 0 to 18 mW. Using a CCD camera and during a preliminary far-field diffraction experiment, the beam waist of the laser output beam was measured equal to 0.45 mm, which is consistent with the manufacturer’s data. The output beam is focused through lens L_1_ of focal length $f_{1}=5.2\;\mathrm{cm}.$ L_1_ defines the *z*-axis, the focal position z=0, $\omega_{0f}=19.6\;µm$ and then $Z_{0}=2.26\;\mathrm{mm}$. The rotation of the two-bladed chopper has been set at a frequency of 10 Hz which gives 50 ms of opening time and 50 ms where the liquid can cool down between two acquisitions. These exposure times provide a sufficient duration for our temporal characterizations because typically, $0.2\;\mathrm{ms}<t_{cz}<5\mathrm{ms}$. The sample under investigations is moved along the beam direction in the focus region. Then, the signal variation of the output diffracted beam is detected on the photodiodes PD1 and PD2, which are connected to a digital storage oscilloscope (DSO). Therefore, the setup allows us to simultaneously record the response of the sample as a function of time and position on the *z* axis. Note that PD2 was used to check that there is no nonlinear absorption or nonlinear scattering in the solution that will be used for the calibration of the experiment. In this way, we are sure that the theoretical model applied here is the right one and does not correspond to a two-photon absorption which should lead to change in the mathematical description. Moreover, this photodiode could be useful to control the signal at the input since there is no nonlinear absorption, especially, to verify that the laser maintains a stable power during the measurement over the z positions.

To illustrate the way to perform the measurements, the time-resolved signal S at different final integration times on the photodiode is shown in Figure 2a. The measurement was made in distilled water containing an absorbent dye so that θ defined by Equation (3b) is around 0.1. It can be seen that S varies significantly at the beginning of the acquisition when t≈0. Whereas, towards the end of the acquisitions when the stationary regime is reached at $t>>t_{c0}$, the response varies slowly. Indeed, for the same difference in interval time 10 ms, (see Figure 2a), the signals in black squares and blue circles show a smaller vertical shift than those between blue circles and red stars. Figure 2b shows the temporal evolution of three z positions chosen on either side of the extremums (V≈±1.7), shown by the stationary black signal ($t>>t_{c0}$). The blue points represent the data and the red lines represent the fits according to Equation (3a). Note that it would be possible to determine the two unknown factors θ and $t_{cz}$ from this fit, but a statistical study has shown a significant fluctuation on the measurements obtained by nonlinear least-squares curve fitting. It is then preferable to proceed step by step, and first determine the value of θ from the black square signal shown in Figure 2a thanks to Equation (5). Next, this value will be injected in Equation (3a) to determine the only unknown which remains, namely $t_{cz}$. Moreover, one might notice that in Figure 2b, the acquisitions and fits do not start at time t=0. This time is difficult to determine exactly, because when the blade of the chopper intrudes the beam, we are in a transition zone where the signal is variable. To reduce this error, it is necessary to reduce the size of the beam and to choose the initial time at the maximum of the derivative between the low and high state of the signal which is roughly situated at the middle of the transition. Thus, the interval that appears between t=0 and the beginning of the signal in Figure 2b relates to the half of this transition zone where Equation (3a) is not valid; therefore, the fitting is carried out using this equation, and translated by a time longer than half of the chopper response (around 0.6 ms in our case). 

Calibration of a measurement system is necessary to take into account the configuration of each experimental setup. Depending on the size of the photodiode, the distance of the photodiode to the focal plane and the size of the beam waist, the experimental conditions can be different. In our case, PD1 which is a $0.8{\;\mathrm{mm}}^{2}$ Si detector with 1 ns rise time was placed at a distance $D+f_{1}=1.45\;m$ from lens L_1_. The calculated far-field diffracted beam waist at this distance was 12.11 mm. Due to the small area of PD1, only 5.5% of the central energy is captured by the photodiode. Using about twenty acquisitions, the system was calibrated with a dye added to distilled water (L=1 mm) with well-known thermo-optical parameters ($dn/dT=-0.8\times 10^{-4}{\;K}^{-1}$; $k=0.61\;{\mathrm{Wm}}^{-1}K^{-1}$) [7]. A commercial solution of brilliant blue food dye FCF (denoted by E number E133) was used. It is a synthetic organic compound employed primarily as a blue colorant, and is highly soluble in water. The measurements are made using an Optical Power Meter (OPM) mounted on the experiment (not shown in Figure 1). This is conducted directly under the same experimental conditions for each acquisition. Otherwise, a spectrophotometer could be used but, based on the same type of photodetector in the visible range (Si), the same results at the same room temperature should be found. So, to facilitate the procedure and because the α dispersion as a function of the wavelength is not needed here, the choice of the OPM is convenient and more practical. So, for each acquisition the absorption was measured owing to the OPM and the TL processing as explained previously. The obtained α values were not equal especially because the equation used for the fitting according to Equation (5) is derived with too many approximations, as is shown in reference [15]. Indeed, in the latter reference, based on the Helmoltz equation, the diffracted intensity profiles at finite distance were calculated numerically without any approximation considering the whole system, including the optical and the thermal parameters. The comparison of the obtained signal S with the one given by Equation (5) shows that even if they have a similar profile, their amplitude can be very different. Taking into account the experimental parameters which are the same as those considered in the simulation of [15], we obtained that the measurement procedure of θ underestimated its value by 60%. Coming back to the OPM measurements, and taking into account the standard deviation σ over the different observations, the results that are shown in Figure 3 give an average value of the absorption: $\alpha_{OPM}=241\pm 14\;m^{-1}$. Then, the calibration was performed so as to obtain the α determined by TL calculations, equal to that of the α measured by the OPM (see Figure 3), considering that the dye is completely dissolved into water and ∅=0. It should be added that the calibration factor found perfect experimental matches to the one predicted by the numerical calculation in [15]; thus, it was necessary to multiply Equation (5) by a factor equal to 0.6 before processing the fit. In Figure 3, we show the dispersion of the measured values $\alpha_{TL}$ obtained with TL fitting versus $\alpha_{OPM}$. The central black star data in this figure shows the mean value calculated over the 20 acquisitions. Note that each $\alpha_{TL}$ measurement is performed on a total of 48 data originating from the Z-profile given by the curve similar to that of the Z-scan as the one shown in black in Figure 2a.

In order to check the linearity of θ for the measuring system, two experiments were performed: (i) in a 1 mm thick cell containing absorbing solution of water ($\alpha =241\;m^{-1}$) by varying P the incident power and by fixing all the other parameters (Figure 4a); (ii) by varying the concentration, and thus the absorption, in two different liquids (Figure 4b): water (blue-stars and red-circles) and methanol (black-squares). 

In Figure 4a we varied P the power and calculated the mean value of θ performing thirty measurements for each black-circular data. The results of the measurement are plotted as a function of the incident power, taking into account the Fresnel reflection on the cell. Thus, 150 measurements are shown under the same experimental conditions. It is observed that the linearity is effective up to about θ=120 mrad, and beyond that, the deviation becomes too crucial. That is why the data obtained for P=2.5 mW is removed from the linear fitting which is represented by the red solid line given by: θ=56.8×P. Theoretically, taking into account all the parameters known and considered to be correct, the latter should be equal to: $$\frac{\theta }{P_{e}}=\frac{L\alpha dn/dT}{\lambda k}=\frac{10^{-3}\times 240\times 8\times 10^{-5}}{532\times 10^{-9}\times 0.61}=59.16{\;W}^{-1}.$$

One can see the excellent precision of the measurement, considering the fitting that crosses 0 with a slope of $56.8{\;W}^{-1}$ to within 3.4% of the value considered as correct. Obviously, this increased precision is due to the high number of points, since here the least squares line is drawn, considering 4×30 measurements. Let us now consider Figure 4b, where the *x*-axis represents the absorption measured with the OPM, while the *y*-axis gives the θ value measured by the fitting (via Equation (5)) considering the used incident power P, λ and L. Soluble dyes were added to make the absorbances large enough to produce easily measurable signals. For very low absorption ($\alpha \leq 50\;m^{-1})$, we divided the concentration by adding known volumes of the solvent to the initially measured one. According to Equation (38) of reference [17]: (6)$$\frac{\lambda \theta }{PL}=\frac{dn/dT}{k}\;\alpha +\varepsilon$$
where ε represents the intersection of the fitting with the *y*-axis, this parameter would in principle allow to trace the absorption of the pure solvent (α=0). We are not going to exploit this propriety in this paper because our distance translation stage is too short. This will be conducted in a future study using a longer Rayleigh range that would allow for thicker cells to be tested and, thus, higher signals from the pure solvent than from the cell walls. The slope of the linear regression lines given by Equation (6) is related to the ratio of the solvent characteristics (dn/dT)/k, regardless of what soluble dye was added to produce the absorbances. An example is given with two different dyes represented by the same slope of the fitting lines corresponding to the blue stars and red circles for water in Figure 4: $\left(dn/dT\right)/k\approx 1.3\times 10^{-4}\;m.W^{-1}$. In addition, we can see the difference in slopes when it comes to methanol: $\left(dn/dT\right)/k\approx 2.1\times 10^{-3}\;m.W^{-1}$, which is one order of magnitude higher.

Another characteristic that can be deduced from this studied setup is the value of k, the thermal conductivity. First, θ is determined by Equation (5) and for each z, the characteristic times $t_{cz}$ is obtained by fitting the transient experimental data (see Figure 2b) using Equation (3a). In Figure 5, $t_{cz}$ is plotted as a function of z where it can be seen that the variation of this time is parabolic as given theoretically by Equation (4). The fitting allows to determine k when c, ρ and ω(z) are known. In the example of Figure 5, the fit was performed with water solution having $\alpha =160\;m^{-1}$. The measured characteristic time of the solution (at z=0) $t_{c0}=0.65\;ms$, from which we can deduce the thermal conductivity coefficient, $k=0.62\;{\mathrm{Wm}}^{-1}K^{-1}$. This value is quite consistent with the assumed value for water $k=0.61\;Wm^{-1}K^{-1}$. 

Subsequently, the feasibility of the measurement was tested to evaluate the scattering induced by gold nanoparticles of a different diameter having a core size contained in intervals of 3–7 nm [22] (maximum of absorption 517 nm) and 47–53 nm [23] (maximum of absorption 537 nm). The suspensions were stabilized in citrate buffer, and the mean hydrodynamic diameters reported by the manufacturer were $D_{h}=11\;\mathrm{nm}$ and $D_{h}=62\;\mathrm{nm}$, respectively. The input power was set high enough so as to obtain a signal characterized by θ≈0.1, and sufficiently low so as not to destroy the electrostatic equilibrium between the nanoparticles and the citrate buffer in order to not induce clusters of gold nanoparticles due to high intensity illumination [24]. The gold nanoparticles of optical density OD=1, measured by the manufacturer with a standard cuvette of 1 cm thickness, were placed in a cell with L=1 mm (OD=0.1) giving $\alpha_{OD}=-\log\left(10^{-0.1}\right)/10^{-3}=230.26\;m^{-1}$. Both suspensions were characterized at 532 nm owing to 20 acquisitions. The representation of the absorption values measured by the TL system is given in Figure 6: blue squares for 5 nm core size and red circles for 50 nm. The central values (stars) show the average measured by the TL method obtaining for the 5 nm nanoparticles $\alpha_{TL}=231.7\pm 8\;m^{-1}$ while for the 50 nm nanoparticles $\alpha_{TL}=183\pm 10\;m^{-1}$. It can be seen that the two measurements given by TL are sufficiently separated and do not overlap, even within the margin of error given by σ. These values allow to calculate ∅, representing the ratio between the number of scattered photons to that of absorbed ones: $\emptyset_{5nm}=-1\pm 9\%$ and $\emptyset_{50\mathrm{nm}}=21\pm 9\%$. Evidently, the obtained values show that the larger particles scatter the light more significantly. 

The feasibility of this method to measure absorption and scattering efficiencies is shown here. The understanding of nanoparticles scattering is very important, especially for the medical area where the majority of imaging techniques and photothermal therapeutic applications are subject to this phenomenon [25]. 

## 4. Conclusions

The objective of this study was to determine the main characteristics of a measurement method using the combined techniques of thermal lens and Z-scan. The experimental calibration has been detailed, corresponding to a correction of the used relationship already numerically provided elsewhere [15]. Then, we focused on the determination of the measurement error from statistical calculations. The linearity of the response was checked, varying either in the incident power or the concentration. The relationship between the slope of this response and the thermo-optical characteristics of the absorbing liquid was determined precisely. It was also demonstrated the possibility of extracting the thermal conductivity from the characteristic time measurements. The results were applied to the determination of the Mie scattering on gold nanoparticles of 5 and 50 nm diameters. 

## Figures and Tables

**Figure 1 materials-15-05008-f001:**
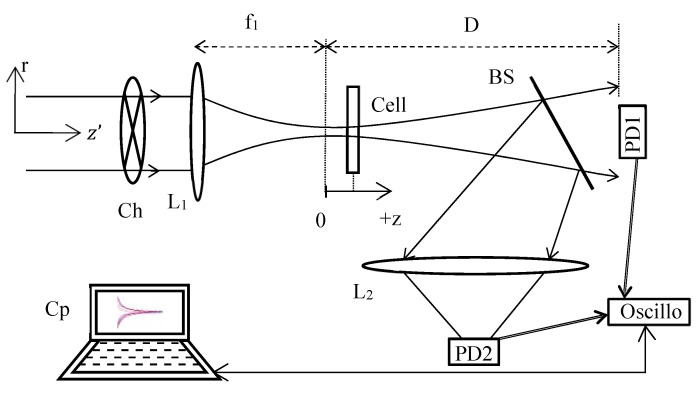
Scheme showing the different position of the optical elements. The cell is scanned along the beam direction around the focal plane (z = 0). The labels refer to: lens (L_1_, L_2_), beam splitter (BS), chopper (Ch), computer (Cp) and photodiodes (PD1, PD2).

**Figure 2 materials-15-05008-f002:**
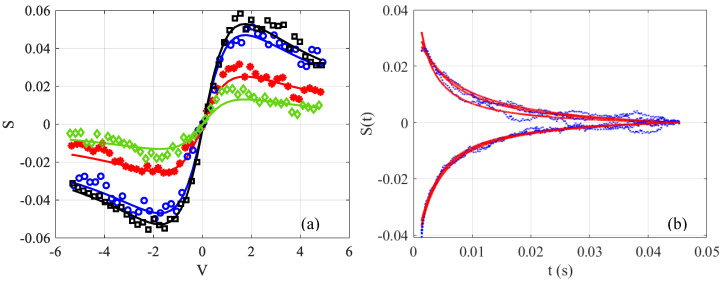
(**a**) Time-resolved Z-scan measurements. Transmittance Z-scan profiles for different final time values: 1.3 ms diamond (green); 3.3 ms stars (red); 13.3 ms circle (blue); and 23.3 ms square (black). Solid lines represent the fittings according to Equation (3a). (**b**) Transient experimental data and corresponding fitting by Equation (3b) for three values of z around the extremums of the signal S in water with ∅=0 (the measured values θ=0.11 and $t_{c0}=0.7\;\mathrm{ms})$.

**Figure 3 materials-15-05008-f003:**
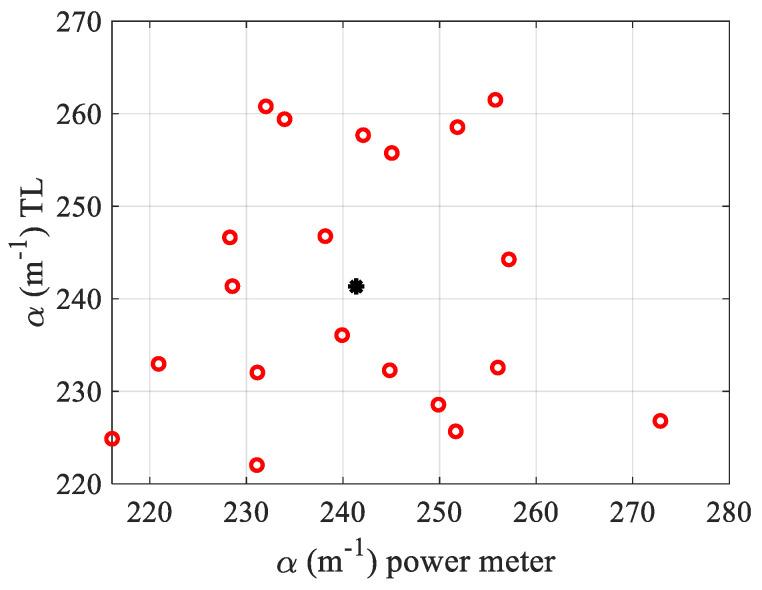
Calibration of the optical setup. The measured absorption through thermal lens process is made equal to the α measured by the OPM through a calibration factor. Empty circles (red): experimental data; star (black): mean value.

**Figure 4 materials-15-05008-f004:**
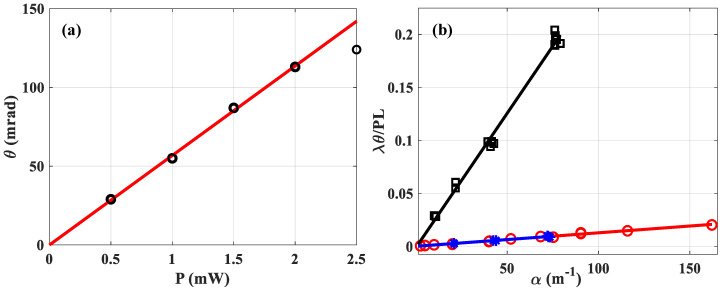
Checking the linearity of the measuring system versus (**a**) the incident power for a fixed absorption, (**b**) the absorption of the solution in different liquids: water (blue stars and red circles) and methanol (black squares).

**Figure 5 materials-15-05008-f005:**
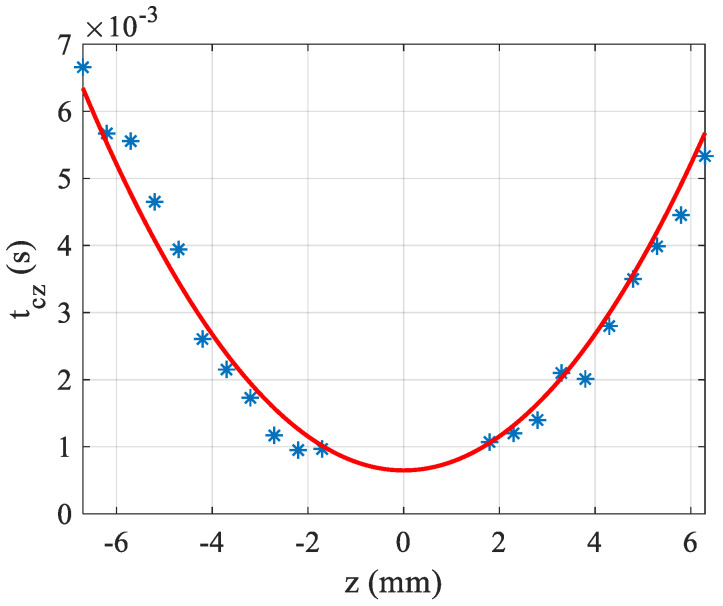
Characteristic time of the solution versus the position of the sample. The data are in blue stars and the solid line is the fitting.

**Figure 6 materials-15-05008-f006:**
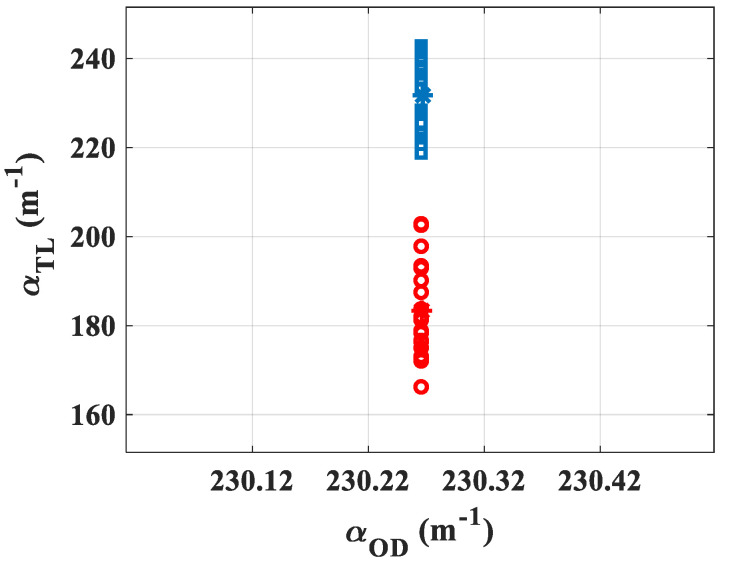
Graphical representation of the absorption values measured by the TL system for OD1 nanoparticles suspensions. The central values (stars) represent the average obtained on $\alpha_{TL}$; blue squares: 5 nm, red circles: 50 nm.

## Data Availability

The data presented in this study are available on request from the corresponding author.

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
