# Peer review of "Time-Resolved cw Thermal Z-scan for Nanoparticles Scattering Evaluation in Liquid Suspension"

_materials, 2022, doi:10.3390/ma15145008_

Round 1

Reviewer 1 Report

The manuscript titled “ Time resolved cw thermal Z-scan for nanoparticles scattering evaluation in liquid suspension The thermal lens effect is analyzed as a time-resolved Z-scan measurements using cw-single Gaussian beam configuration

In this manuscript, the topic is interesting and very motivating.  

I have some notes regarding this manuscript:

1-    The references are written in italic letters.

2-      Could we measure the nonlinear optical parameters such as nonlinear absorption and nonlinear refraction coefficients simultaneously with thermos-optic coefficient?

3-     What are the situation if we used nanosecond laser instead of CW?

Author Response

See uploaded file

Reviewer 2 Report

This manuscript reports on thermal lens effects in aqueous dye solutions and gold nanoparticle suspensions measured by a cw-single beam time-resolved Z-scan method. The calibration formula to evaluate the absorption coefficient from the Z-scan signal is obtained from the results of dye aqueous solutions, and the ratio of scattering efficiency to absorption efficiency for gold nanoparticles is evaluated using the calibration formula. However, there are several points that need to be addressed in order to be published in Materials, as follows.

[1] p.2 ? absorbance in ?−1  ? absorption coefficient

[2] p.5 The measurement was made in distilled water containing an absorbent dye so that ? defined by Eq. 3(b) is around 0.1.

What dye was used in what concentration? It is strange that this information is not given; from this, the α at 532 nm should be more precisely known than measured with OPM described below and it can be checked if the photothermal lens method works well and if the results in agreement with the theoretical formula are obtained in Fig.2 and Fig.3.

p.5 For each acquisition the absorption was measured owing to an Optical Power Meter (OPM) and the TL processing as explained previously. The obtained α values were not equal especially because the equation used for the fitting according to Eq. (5) is derived with too many approximations as it was shown in reference [xv]. Taking into account the standard deviation σ over the different observations, the results that are shown in Fig. 3, give an average value of the absorption: ????=241±14 ?−1.

[3] p.7 (??/??)/?[ W-1]    The unit looks incorrect because ? thermal conductivity in ?/?/?/?(W/m/K) 

[4] p.8 The central values (stars) show the average measured by both methods OPM and TL obtaining for the 5 ?? nanoparticles ????=224±6 ?−1 and ???=203±4 ?−1 while for the 50 ?? nanoparticles ????=232±8 ?−1 and ???=182±10 ?−1. It can be seen that the two measurements given by TL are sufficiently separated and do not overlap even within the margin of error given by σ. These values allow to calculate representing the ratio between the number of scattered photons to that of absorbed ones: 5??=10±5 % and 50??=21±9 %. Evidently, the obtained values show that the larger particles scatter the light more significantly.

 Firstly, it is not clear to the reviewer how the values of ∅5??=10±5 % and ∅50??=21±9 % are obtained. Secondly, since the size of gold nanoparticles is less than 1/10 of the laser wavelength of 532 nm, the scattering intensity efficiency should nearly follow (size)^6 rule. From this rule, the ratio of ∅5??=10±5 % with respect to ∅50??=21±9 % is definitely too large (∅5?? should be much smaller), although the size order of ∅ is qualitatively appropriate. These values raise the question whether the scattering efficiency is accurately measured. The most important result of this paper is that the scattering efficiency can be evaluated, so the authors need to provide enough explanation to convince the reader about these.

Author Response

see uploaded file

Round 2

Reviewer 2 Report

In the revised manuscript, [1] and [2] are satisfactorily addressed, but [3] and [4] are not.

[3] p. 12 in the revised: “In addition, we can see the difference in slopes when it comes to methanol: (????)?≈2.1×10−3 ?−1 which is one order of magnitude higher.”  The unit is not corrected.

[4] The values have been updated to “5??=−1±9 % and 50??=21±9 %.” in the revised from “5?? =10±5 % and 50?? =21±9 %in the original. The experimental data on which these values are based have also been updated as follows.

In the revised:

???=231.7±8 ?−1 for the 5 ?? nanoparticles

???=183±10 ?−1 for the 50 ?? nanoparticles

with ???1=230.25 ?−1

5??=−1±9 % and 50??=21±9 %

 In the original:

????=224±6 ?−1 and ???=203±4 ?−1 for the 5 ?? nanoparticles

????=232±8 ?−1 and ???=182±10 ?−1 for the 50 ?? nanoparticles.

5?? =10±5 % and 50?? =21±9 %.

Certainly, with these data, the values are within a reasonable range for the size of the particles, but why have the data been updated so dramatically? The authors need to explain. In addition, the values on the horizontal axis in Fig. 6 are strange.

Round 3

Reviewer 2 Report

In the revised version after 2nd round review, all the concerns are satisfactorily addressed, so that the paper can now be published.

 Please delete the duplication of Fig. 6.